# Social foraging in vampire bats is predicted by long-term cooperative relationships

**Simon P. Ripperger** [1,2,3]*, **Gerald G. Carter**[1,2]*

**1** Department of Evolution, Ecology and Organismal Biology, The Ohio State University, Columbus, Ohio, United States of America, **2** Smithsonian Tropical Research Institute, Balboa, Ancón, Panamá, **3** Museum für Naturkunde, Leibniz-Institute for Evolution and Biodiversity Science, Berlin, Germany

* simon.ripperger@gmail.com (SPR); carter.1640@osu.edu (GGC)

## Abstract

Stable social bonds in group-living animals can provide greater access to food. A striking example is that female vampire bats often regurgitate blood to socially bonded kin and non-kin that failed in their nightly hunt. Food-sharing relationships form via preferred associations and social grooming within roosts. However, it remains unclear whether these cooperative relationships extend beyond the roost. To evaluate if long-term cooperative relationships in vampire bats play a role in foraging, we tested if foraging encounters measured by proximity sensors could be explained by wild roosting proximity, kinship, or rates of co-feeding, social grooming, and food sharing during 21 months in captivity. We assessed evidence for 6 hypothetical scenarios of social foraging, ranging from individual to collective hunting. We found that closely bonded female vampire bats departed their roost separately, but often reunited far outside the roost. Repeating foraging encounters were predicted by within-roost association and histories of cooperation in captivity, even when accounting for kinship. Foraging bats demonstrated both affiliative and competitive interactions with different social calls linked to each interaction type. We suggest that social foraging could have implications for social evolution if "local" within-roost cooperation and "global" outside-roost competition enhances fitness interdependence between frequent roostmates.

## Introduction

Socializing and foraging are 2 key determinants of reproduction and survival that can influence each other in several interesting ways. Preferred social relationships can drive foraging decisions (e.g., great tits [1]). Conversely, shared foraging behaviors might shape how relationships form (e.g., bottlenose dolphins [2]). Social relationships can determine access to food because closely affiliated individuals can peacefully co-feed at a food patch, hunt together [3], cooperatively defend food patches (e.g., [4–6]), or even give food to less successful foragers (e.g., chimpanzees [7]). Access to food is therefore one benefit of long-term cooperative relationships, i.e., stable preferred associations that involve cooperative investments such as grooming and food sharing. For example, grooming in chacma baboons promotes tolerance during foraging [8], and vervet monkeys strategically groom individuals that control access to

**Data Availability Statement:** All relevant data and R-code underlying are available from figshare. Ripperger, Simon; Carter, Gerald (2021): Data and R code for "Social foraging in vampire bats is

predicted by long-term cooperative relationships":
https://doi.org/10.6084/m9.figshare.14529588.v2.

**Funding:** This study was funded by grants of the
National Science Foundation (Integrative
Organismal Systems #2015928; GGC; https://
www.nsf.org/), of the Deutsche
Forschungsgemeinschaft (https://www.dfg.de/)
within the research unit FOR-1508, a Smithsonian
Scholarly Studies Awards grant (GGC, SPR; https://
www.si.edu/), and a National Geographic Society
Research Grant WW-057R-17 (GGC; https://www.
nationalgeographic.com/). The funders had no role
in study design, data collection and analysis,
decision to publish, or preparation of the
manuscript.

**Competing interests:** The authors have declared
that no competing interests exist.

**Abbreviations:** GPS, Global Positioning System;
ICC, intraclass correlation coefficient; IR, infrared;
LMM, linear mixed-effect model; MRQAP, multiple
regression quadratic assignment procedure with
double semi-partialling; NSF, National Science
Foundation; PHS, Public Health Service; QAP,
quadratic assignment procedure.

food due to social dominance [9] or an experimentally manipulated ability to access food [10].
A particularly clear nonprimate example of cooperative relationships providing food occurs in
common vampire bats where females regurgitate ingested blood to socially bonded kin and
nonkin that failed to feed that night [11,12].

Food-sharing relationships in vampire bats form as preferred associates escalate social
grooming [13]. These preferred associations and cooperative interactions occur within the day
roost. However, little is known about if or how cooperative relationships extend beyond the
roost. For example, foraging with socially bonded roostmates might increase efficiency in
searching for prey or feeding from wounds, but it remains unclear if or how vampire bats per-
form social hunting. Several authors provide anecdotal evidence for groups of females appar-
ently flying together, adult females departing roosts in groups of 2 to 6, and groups arriving
together at a pasture or approaching and circling prey [14–17]. There are also observations of
up to 4 individuals feeding simultaneously from different wounds on the same cow [14] or
pairs feeding on the same wound [14,16]. Wilkinson [16] described evidence that mother–
daughter pairs co-forage and share wounds, but found no evidence that frequent roostmates
forage together.

Social foraging can take many forms, from mere aggregations of individuals attracted to a
common resource to coordinated foraging groups with differentiated roles. Socially hunting
species can be placed on a spectrum of resource sharing from individual foragers competing to
group-level sharing [3]. The form of social foraging and the scale of competition over
resources outside the roost can have implications for the evolution of food-sharing relation-
ships. Several evolutionary models of vampire bat food sharing as multilevel selection view
them as foraging individually then sharing food at the group level [18–20], but this view con-
trasts with evidence that food-sharing relationships within groups are reciprocal and highly
differentiated [11,21]. An alternative possibility is that individualized relationships drive both
within-roost resource sharing and social hunting. This hypothesis is not mutually exclusive
with group hunting, because even if individuals forage in groups, specific pairs could be more
likely to compete or share a wound or host [14–16,22].

Here, we assessed the relative evidence for a range of hypothetical scenarios that vary in
degree of coordination of social foraging among socially bonded bats (Fig 1). In the simplest
case, preferred roostmates might not coordinate their behavior outside the roost. If instead
bats optimize individual foraging efficiency by preferentially departing, following, or foraging
with their preferred roostmates, then within-roost networks should predict co-departures or
foraging encounters. Alternatively, to maximize their collective search area, bats might prefer
to forage with bats outside their network of cooperative relationships and actually avoid forag-
ing with their frequent roostmates. If so, within-roost and outside-roost networks should be
negatively correlated. Finally, if entire roosting groups forage together, then we expect simi-
larly dense and highly correlated within-roost and outside-roost networks.

To evaluate evidence for these scenarios, we tested whether nightly foraging departures and
encounters were predicted by kinship, roosting associations based on 2 levels of proximity
(during the previous day or over the whole study), and rates of social grooming, food sharing,
and co-feeding in captivity. To document roosting associations and foraging encounters, we
analyzed social encounter data from proximity sensors placed on 50 free-ranging common
vampire bats. As additional predictors for 23 of these bats, we used published long-term rates
of social grooming and food sharing [23] and co-feeding rates from when these bats were cap-
tive. Using simultaneous ultrasonic recording and infrared (IR) video, we also describe a dis-
tinct new type of vampire bat call only observed during hunting interactions. Multiple lines of
evidence show that cooperative relationships in vampire bats extend outside the roost. More

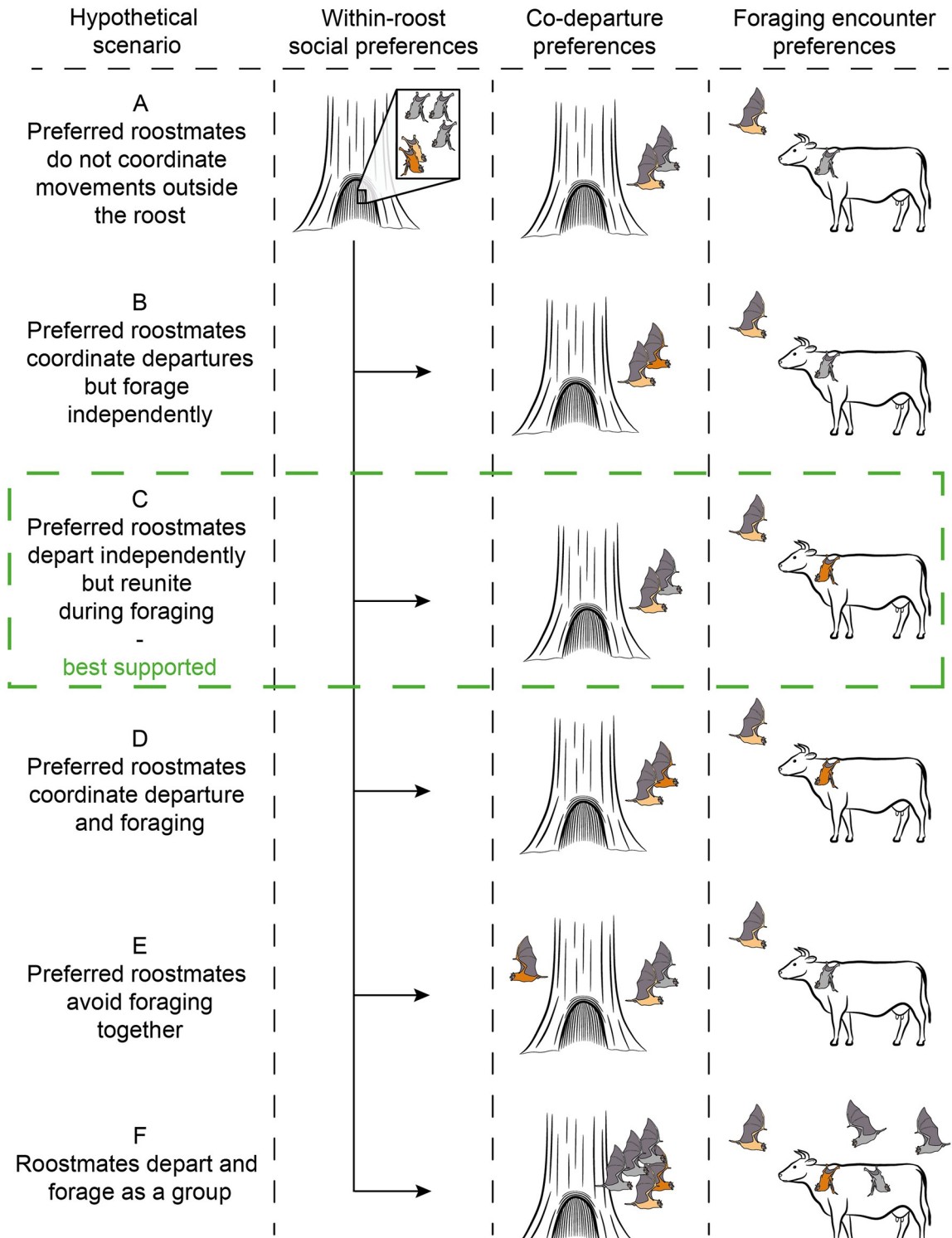

**Fig 1. Hypothetical scenarios for how within-roost relationships could predict foraging.** For the same roosting association networks, each scenario predicts different outcomes for how preferred within-roost relationships correlate with co-departures or encounters during foraging. Preferred roostmates (shown as pair of light brown and dark brown bats) might either **(A)** not coordinate their behavior outside the roost, **(B)** coordinate only their departures, **(C)** depart independently and reunite during foraging, **(D)** coordinate departures and foraging, or **(E)** avoid foraging together. Alternatively, the bats could **(F)** depart and forage as a large group.

generally, our findings illustrate how social relationships and networks can both extend and vary across contexts.

## Methods

### Subjects

Subjects were common vampire bats (*Desmodus rotundus*) including 27 wild-caught adult females that were tagged and released and 23 previously captive females (17 adults and their 6 subadult captive-born daughters) that had spent the past 21 months in captivity and were then tagged and released back into their wild roost tree (see [13,23]). See S1 Text for details.

### Kinship

We assumed that known mother–daughter pairs had a kinship of 0.5. To estimate kinship for all other pairs, we genotyped bats at 17 polymorphic microsatellite loci (DNA isolated via a salt–chloroform procedure from 3- to 4-mm biopsy punch stored in 80 or 95% ethanol), then used the Wang estimator in the R package "related." See S1 Text for details.

### Past cooperative interaction rates in previously captive bats

To measure cooperative relationships in the previously captive bats, we used previously published rates of social grooming and food sharing from experimental fasting trials [13]. See S1 Text for details. To assess tolerance while feeding, we also analyzed observations of co-feeding among the same captive vampire bats. Social interactions were observed at blood spout feeders while the bats were in captivity, including 1,300 competitive interactions and 277 cases of co-feeding where 2 bats were observed feeding from the same blood spout at the same time (from 1,050 hours of observation from 70 nights). In 201 of these cases, both bats were clearly identified. We used these to construct a co-feeding network of the number of dyadic co-feeding events (range = 0 to 6) for each pair.

 To test correlations between the captive co-feeding network and networks of food sharing or social grooming, we used Mantel tests. To test the same correlation while controlling for overlap in individual feeding times, we also used a custom double permutation test [24]. This procedure calculates an adjusted co-feeding rate for each pair as the difference between the observed co-feeding rate and the median expected co-feeding rate from 5,000 permutations of the co-feeding bat identities, permuted among the bats seen within each hour. To test for preferred captive co-feeding partners, we also used the same within-hour permutations to test if social differentiation in co-feeding (the coefficient of variation in co-feeding rates) was greater than expected from the null model.

### Association rates in the wild using proximity sensors

We placed custom-made proximity sensors on all 50 female common vampire bats (sensor mass: 1.8 g; 4.5% to 6.9% of each bat's pre-feeding mass) that automatically documented dyadic associations among all 50 tagged bats when those come within the reception range of 5 to 10 m [23,25]. To log encounters, each proximity sensor broadcasted a signal every 2 seconds to update the duration of each encounter. We used 1 second as the duration of encounters that were shorter than 2 successive signals (i.e., encounters shorter than 2 seconds). The maximum signal strength of each encounter is used as an estimate for a minimum proximity between 2 tagged bats during the encounter by comparing the signal intensity to a calibration curve [23,25,26].

We collected association data on the free-ranging bats at Tolé, Panama (8° 12′ 03″ N 81° 43′ 46″ W), a rural area that is mainly composed of cattle pastures for meat production. Around 200 to 250 common vampire bats roosted inside a hollow tree on a cattle pasture that was about 15 ha in size. To create a stable food patch during part of our study, we corralled approximately 100 heads of cattle at a distance of approximately 300 m from the roost from 6 PM until 6 AM between the evening of September 21 until the morning of September 26, 2017 (days 1 to 5 in our study). Before and after that time period, the cattle were ranging freely. A neighboring, much larger pasture west of the roost had about 1,500 heads of cattle within a distance of 1 to 2 km (Fig A in S1 Text).

To construct networks of roosting association rates during each daytime period within the roost, we relied on roosting association data that had been used in a previous study [23]. Based on the same 2 thresholds of signal strength as before, we defined 2 categories of proximity: "associations" (within approximately 50 cm) and "close contacts" (within approximately 2 cm). Roosting network edges were rates of within-roost association or close contact, i.e., the total time 2 bats spent in association per unit of time. See S1 Text for details.

To help localize bats, we used base stations that can detect tagged bats at distances of about 150 m. We placed these stations at the roost and at 5 other locations in the surrounding cattle pastures to help localize individuals and encounters as inside or outside the roost. To identify departures from the roost, we found the points in time where each bat lost connection from the roost base station and almost all of the many tagged bats in the colony within communication range (i.e., a sudden drop in associations from many bats down to 0 to 3 bats; see [25]). Some departing bats also contacted base stations on the cattle pasture (Fig A in S1 Text). We used the same kind of data to infer the return times to the roost for each bat and night.

Of the 629 dyadic encounters that occurred 1 minute after leaving the roost and 1 minute before arriving at the roost, we excluded 43 encounters from further analysis, because a proximity sensor contacted the roost base station, suggesting that those encounters occurred while bats were roosting at the entrance or on the outside of the roost tree. The remaining 586 encounters occurred farther away, outside the communication range of the roost base station, and we refer to these as "foraging encounters."

## Observing interactions of foraging vampire bats

At Tolé, we only observed 2 occasions where 2 bats stopped at the monitored cattle pasture and were associated (for 3.5 and 4.6 minutes). When releasing the corralled cattle in the morning, we observed bite marks. However, to avoid changing their behavior, we did not get close enough to the cattle at night to record audio or video of bats interacting. To collect direct observations on foraging behavior, we therefore recorded simultaneous audio and video of bat foraging behavior at a different farm near La Chorrera, Panama (8° 52′ 42″ N 79° 52′ 05″ W) using an IR spotlight, IR-sensitive video camera (Sony AX53 4K camcorder), and an Avisoft condenser microphone (CM16, frequency range 1 to 200 kHz) and digitizer (Avisoft USG 116 Hbm, 1,000 kHz sampling rate, 16-bit resolution) connected to a notebook computer. One observer (SPR) moved with a herd of about 20 grazing cattle without visible light, only using the viewfinder of the IR camera. To compare social calls made during foraging with calls from inside a roost, we used the same recording equipment to record social calls from a roost only a few hundred meters from the foraging site at La Chorrera.

## Acoustic analysis of calls in foraging bats

We used Avisoft SASLab Pro (Raimund Specht, Avisoft Bioacoustics, Glienicke/Nordbahn, Germany; version 5.2.13) to measure acoustic parameters of the social call types. Start and end

of calls were determined manually, based on the oscillogram. Subsequently, 5 acoustic parameters were measured automatically: 1 temporal (duration) and 4 spectral parameters (peak frequency at maximum amplitude, minimum and maximum frequency, and bandwidth). Acoustic parameter extraction was restricted to the fundamental frequency. Spectrograms were created using a Hamming window with 1024-point fast Fourier transform and 93.75% overlap (resulting in a 977 Hz frequency resolution and a time resolution of 0.064 ms). To estimate the frequency curvatures of the different call types, we measured the spectral parameters at 11 different locations distributed evenly over the fundamental frequency of each call. To compare call structure from different contexts (roosting versus foraging and antagonistic versus affiliative behavior) in multivariate space, we plotted the first 2 principal components after entering these measures into a principal component analyses with varimax rotation (using the "foreign" package in R).

## Statistical analysis of foraging behavior

To estimate foraging bouts, we calculated the periods when each bat was distant from the roost tree (S1 Fig). Then, to test whether the previously captive bats and never-captive control bats differed in their departure times and foraging bout durations, we fit linear mixed-effect models (LMMs) with type of bat and day as fixed effects and bat as a random intercept. We calculated *p*-values using Satterthwaite degrees of freedom method with the R package lmerTest. To compare consistency of onsets and durations, we measured the unadjusted repeatability (intraclass correlation coefficient or ICC) for each type of bat. To count how often tagged bats departed together, we counted and inspected cases where bats departed the roost within 1 minute of each other.

## Preferred associations during foraging

To test if repeated foraging encounters occurred among the same bats more than expected by chance, we used a permutation test that compared observed and expected social differentiation while controlling for overlap in foraging times. For social differentiation, we used the coefficient of variation in co-foraging rates, which increases when some pairs have more repeated encounters than others and decreases when all pairs have similar encounter rates. We first used a simple and conservative measure of co-foraging: counting the presence or absence of an encounter during each hour outside the roost over 9 days. These counts varied from 0 to 15. If 2 bats met twice in the same hour, this is still one encounter. We used this method because all bats were sampled evenly within each night and most foraging encounters were brief (median = 1 second). These present versus absent observations in each hour were swapped to randomize the data. Specifically, we permuted one bat in every dyad to a random possible partner that was also outside the roost during that same day and hour. By randomizing the data this way 5,000 times, we generated a null distribution of social differentiation values expected by chance.

## Predictors of social foraging

To test predictors of social foraging, we constructed foraging encounter networks where edges were based on either duration of total encounter time outside the roost (seconds) or number of days with foraging encounters (0 to 9 days). The latter response variable is far more conservative because it only counts repeats across different days. We included the following predictors: kinship, within-roost association rate, within-roost close contact rate, social-grooming rate, and food-sharing rate. We also tested the effect of dyad type (i.e., both bats previously captive, both bats never captive, one bat previously captive, or both bats captive-born

juveniles). We did not use number of nights with foraging encounters as a response for tests that only included the previously captive bats, because 9 of these bats (including all captive-born bats) left the roost during the study period [23].

To test the effect of predictor networks on a response network, we used regression quadratic assignment procedure (QAP) for single predictors or multiple regression quadratic assignment procedure with double semi-partialling (MRQAP) for 2 predictors (using the "asnipe" R package [27]). To create null models, we used constrained (within-day) node label permutations. This approach is necessary for preserving the daily and nightly network structure (e.g., distribution of edges and edge weights) and for controlling for the presence or absence of bats in the roost each day. To control for foraging bout overlap in each pair, we included that measure as a covariate. We also used QAP to test whether the within-roosting association on each day predicted the subsequent foraging network that night. We then bootstrapped the mean of the slopes across the 8 days to test for an overall paired day–night effect.

## Consistency of individual social traits

To test whether bats that are more socially connected within the roost are also more connected in foraging networks, we tested if the nodes' degree centrality was correlated between roosting and foraging networks. We measured degree centrality independently within each day or night network when the bat was present and then took the mean for each bat. Bats with no encounters in that day or night were considered missing for that day (i.e., not counted as zero degree). We fit general linear mixed effect models with foraging network centrality as the response variable, roosting network centrality (either association and close contact) as fixed effect, and bat as random intercept. p-Values were calculated from 5,000 permutations of the bat's foraging centralities within each night (i.e., constrained node label permutations (within night) control for the fact that foraging and roosting network centralities could be correlated simply by some bats being present at the site longer). Throughout the results, we use "p-null" to indicate p-values that come from a null model where permutations were constrained within day. All data and R code are available on Figshare [28].

## Ethics statement

Our protocols adhered to the following guidelines: (1) The US Government Principles for the Utilization and Care of Vertebrate Animals Used in Testing, Research, and Training, developed by the Interagency Research Animal Committee and adopted in 1985 by the Office of Science and Technology Policy; (2) The Animal Welfare Act, 7 United States Code (USC) §2131 et. seq., and the regulations promulgated thereunder by the US Department of Agriculture (USDA); and (3) Public Health Service (PHS) Policy on Humane Care and the Use of Laboratory Animals, August 2002, for all PHS- or National Science Foundation (NSF)-supported activities involving vertebrate animals. All experiments were approved by the Smithsonian Tropical Research Institute Animal Care and Use Committee (#2015-0915-2018-A9 and #2017-0102-2020) and by the Panamanian Ministry of the Environment (#SE/A-76-16 and #SE/AH-2-17).

## Results

### Sampled bats did not depart together

The never-captive control bats departed from the roost 8.3 hours after sunset and returned 2.5 hours later, on average (S1 Fig). The previously captive bats foraged earlier and less predictably (see below). We observed only 5 cases where 2 bats departed within 5 seconds of each

other, and none of these cases was followed by a foraging encounter. For the cases where pairs did have a foraging encounter, the shortest differences in departure times were 8, 21, and 28 seconds.

### Previous captivity influenced departures and foraging

Compared to the never-captive control bats, the previously captive bats departed the roost on average 1.6 hours earlier (LMM, t = −4.55, $p < 0.0001$), but they did not forage consistently longer (t = 1.29, $p = 0.2$; S6 Data). The captive-born bats departed 2 hours earlier (t = −3.15, $p = 0.002$) and also did not forage longer (t = −0.41, df = 47.8, p = 0.7) than control bats. All these models control for departure times being on average 14 minutes later each day (t = 6.6, $p < 0.0001$; S6 Data), perhaps due to moonset times being approximately 40 to 45 minutes later each day during the study period. The total duration of foraging encounters did not clearly differ between types of pairs (S3 Data, left), but pairs of control bats had significantly more nights with foraging encounters (S3 Data, right) compared to other types of pairs, possibly due to control bats having more consistent foraging times. Departure times were more consistent across days within each control bat (ICC = 0.58) compared to within each previously captive bat (ICC = 0.21) or captive-born bat (ICC = 0). The duration of the longest foraging bout was also more consistent in wild control bats (ICC = 0.54) than in the previously captive bats (ICC = 0.35) or captive-born bats (ICC = 0.15).

### Preferred associations in foraging encounter networks

Foraging encounters were orders of magnitude shorter in duration than within-roost encounters; their median duration was 1 second, and they never exceeded 30 minutes (S2 Data). Of 151 pairs with a foraging encounter, 45 did this repeatedly across 9 nights. Pairs of bats varied in the number of hours in which they reunited, and this variation was greater than expected from our null model that simulated random encounters among bats that were outside the roost in the same hour (observed social differentiation = 4.36; p-null < 0.001; 95% of expected values: −2.2 to 2.4). Most of these foraging encounters occurred at locations outside our sampled areas, but 10 encounters (involving 8 pairs of bats) occurred near the other base stations on the surrounding cattle pastures (Fig A in S1 Text), and only 3 foraging encounters (among 3 pairs) occurred at the corral that we created as a stable food patch about 300 m from the roost (2 encounters on days 1 and 3 while the cattle were present and 1 encounter on day 7).

### Kinship predicts foraging encounters

Kinship predicted the number of nights with foraging encounters (QAP, β = 15.4, *n* = 46 bats, p-null < 0.0001) and foraging encounter time (β = 15.4, *n* = 47 bats, p-null = 0.022) even when controlling for bout overlap (MRQAP, β = 0.10, $p = 0.002$; Fig 2). The median duration of a foraging encounter for close kin (kinship >0.1) was 9 seconds, compared to 1 second for non-kin (kinship <0.1).

### Within-roost association rates predicted foraging encounters

Bats that spent more time near each other within the tree during the day also spent more time together outside the roost during the night (associations: QAP, β = 29.5, p-null < 0.001; close contact: QAP, β = 24.7, p-null = 0.002) even when controlling for the foraging bout overlap (associations: MRQAP, β = 0.092, $p = 0.003$; close contact associations: MRQAP, β = 0.078, $p = 0.015$). Pairs with greater within-roost association rates also had foraging encounters on

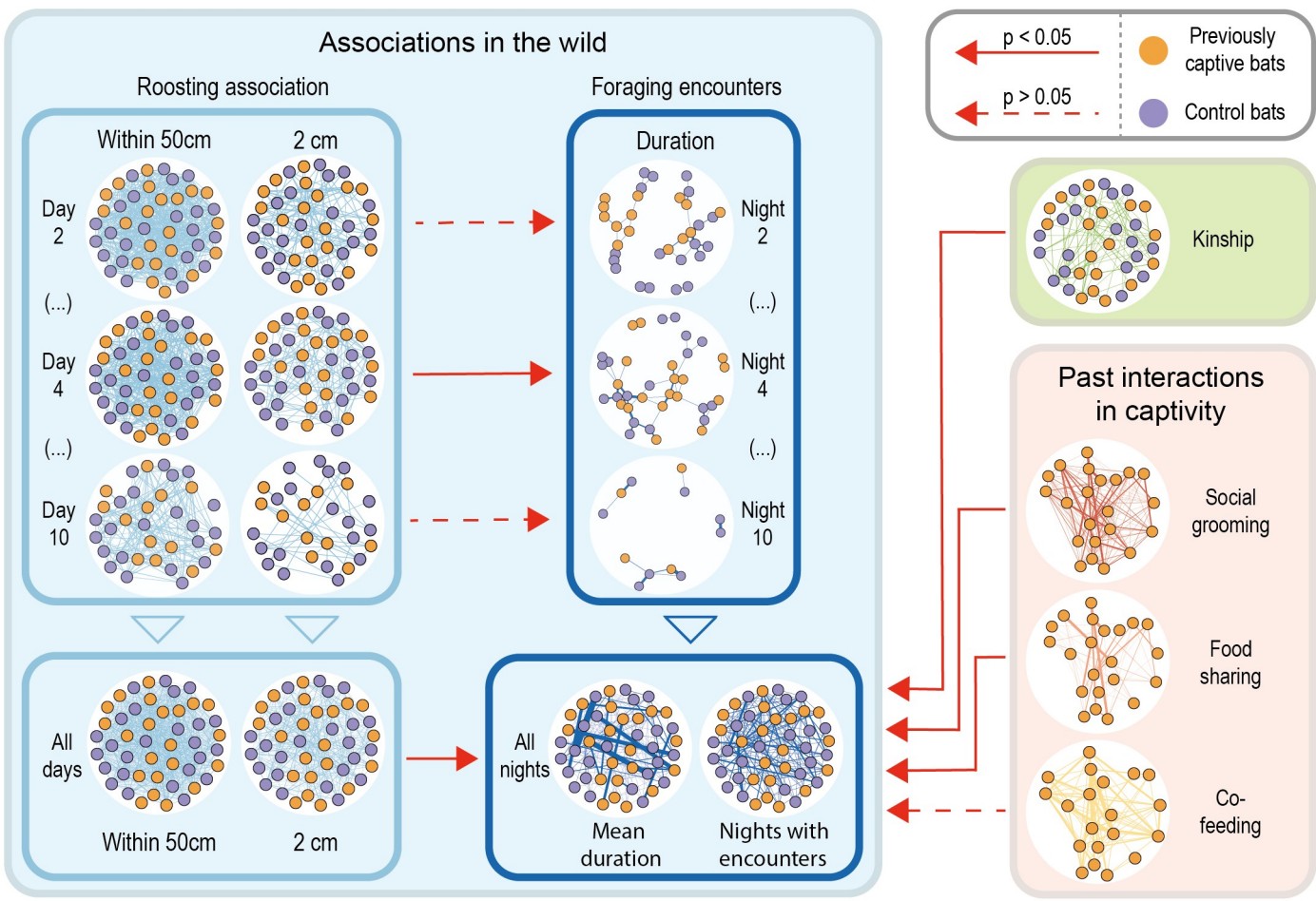

**Fig 2. Network comparisons.** Foraging encounter rates were predicted by roosting associations, kinship, and previous long-term rates of social grooming and food sharing in captivity. To facilitate visual comparisons, we applied the Fruchterman–Reingold algorithm to a fully connected unweighted network and used this layout to fix the spatial coordinates of nodes across networks (except for the sparse night-by-night foraging networks), we scaled edge strength in each network, and we removed nodes without edges. In the kinship network, only edges with kinship estimates >0.24 are shown, and bats without kin in the group are not plotted. In the paired night–day networks of association in the wild, we only detected a clear correlation between day and night networks on day 4 (Table A in S1 Text). Numerical values underlying this figure are available in S1 Data.

more nights (associations: QAP, β = 0.07, p-null < 0.001; close contact association: QAP, β = 0.04, p-null = 0.021; Fig 2).

When we tested the effect of each pair's daytime roosting proximity on their foraging encounters on that subsequent night, we found a clear effect within only 1 of the 8 days (Table A in S1 Text), but the overall effect size across days was greater than 0 (associations: mean β = 0.026, 95% CI = 0.004 to 0.051; close contact associations: mean β = 0.018, 95% CI = 0.003 to 0.04).

## Roosting degree centrality predicted foraging degree centrality

Bats that associated with more partners within the roost also associated with more partners at night outside the roost (associations: β = 0.034, n = 48 bats, one-tailed p-null = 0.008; close contact: β = 0.055, one-tailed p-null = 0.078; S4 Data). *p*-Values are one-tailed because the center of the expected β values from the null model was not 0 (S4 Data).

## Cooperative relationships in captivity predict foraging encounters in the field

In the previously captive bats, kinship and cooperative relationship were independent predictors of social foraging. Foraging encounter time was predicted by food sharing (β = 38.7, *n* = 22, p-null = 0.015; MRQAP controlling for bout overlap: β = 0.20, *p* = 0.014), by food sharing when controlling for kinship (MRQAP, sharing: β = 0.16, *p* = 0.022; kinship: β = 0.14, *p* = 0.049), and by social grooming (QAP, β = 26.5, *n* = 22, p-null = 0.032), but the effect of social grooming was unclear when controlling for bout overlap (MRQAP, β = 0.12, p-null = 0.063).

## Co-feeding among familiar captive bats was not limited to cooperative relationships

In contrast to the evidence for social differentiation in the field, we detected only weak evidence for preferred associations during co-feeding in captivity (social differentiation = 2.10, p-null = 0.047 when controlling for hour, p-null = 0.041 when not controlling for hour), and we found no correlation between captive co-feeding and social grooming, food sharing, or social foraging time in the wild (see Table B in S1 Text).

## Behavioral interactions during foraging

To sample bat interactions during foraging encounters, we recorded IR video and ultrasonic audio of 14 interactions between foraging vampire bats (Tables B and C in S1 Text). Social calls during foraging had 3 general spectral shapes (Fig 3, S5 Data). "Downward sweeping calls" are also recorded often in roosts (Fig E in S1 Text) and are produced by socially isolated vampire bats in captivity [29,30]. "Buzz calls" were noisy without clear tonal structure and occurred during antagonistic interactions. We observed "n-shaped calls" produced by bats interacting while near cattle (Fig 3). To our knowledge, this call type is distinct from others (S5 Data) and has never been seen in wild roosts, from confrontations at the feeders in captivity [31], or from individually isolated bats in captivity [29].

## Discussion

### Long-term cooperative relationships predicted repeated foraging encounters

All the female vampire bats we tagged departed the roost separately, but they often reunited far from the roost during foraging bouts (Fig 1). The rates of these foraging encounters were consistently higher than expected in specific pairs. These frequent encounters were predicted by roosting associations, kinship, and the history of social grooming and food sharing in captivity, even when accounting for kinship. Previous experiments with female vampire bats suggest that these measures—roosting proximity, social grooming, and food sharing—reflect an underlying cooperative relationship [11–13,16,23]. Here, we knew the cooperation histories among the previously captive bats and that these individuals had no interactions with the control bats for at least the previous 21 months. We could therefore infer that relationships typically defined by associations and cooperative interactions within roosts, also extend beyond the roost and may provide benefits during foraging (see "Implications for cooperation"). In addition to consistent social relationships across context (from captivity to roosting to foraging), we found evidence that bats that encountered more associates in the roost during the days also encountered more associates while foraging during the nights, suggesting consistent

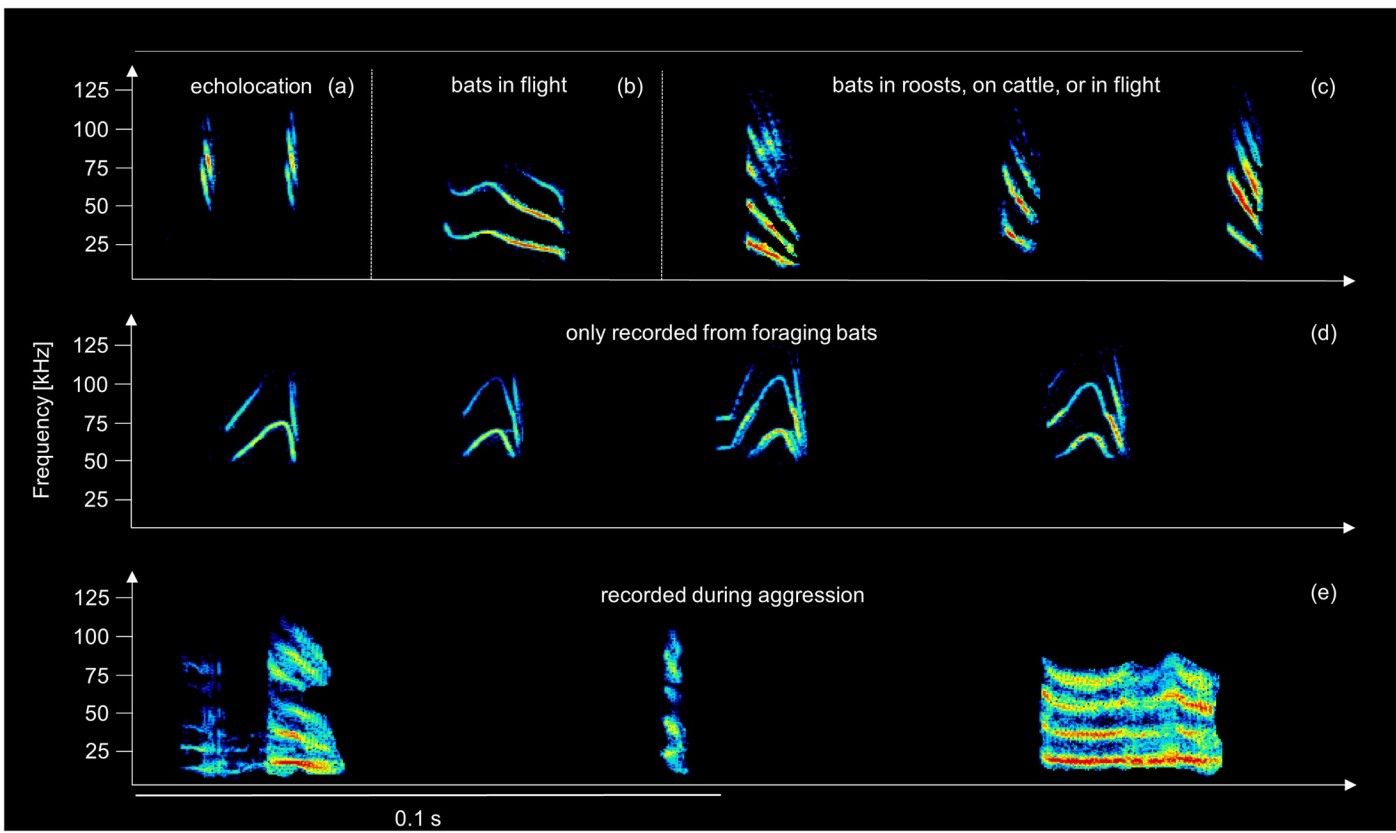

**Fig 3. Spectrograms of social calls of common vampire bats flying near or attacking free-ranging cattle on a pasture.** Behavioral context was derived from synchronized video, and we identified the calling bat when mouth movements were visible. Calls include **(a)** echolocation calls (biosonar), **(b)** undulated down sweep, which was only observed in one recording where 2 bats where flying near a cow, **(c)** down sweep calls, **(d)** n-shaped calls, and **(e)** buzz calls recorded while 2 bats engaged in antagonistic behavior on a single cow. Inter-call intervals were modified for the figure, except for the call sequence in panel (e), a sequence recorded from 2 aggressively interacting bats.

individual variation in social traits. Taken together, these observations of social foraging cannot be fully explained by nonsocial factors such as shared site preferences.

Although some foraging encounters may have occurred before or after foraging, most of these encounters were likely to have occurred during foraging for several reasons. First, foraging encounters were brief, whereas associations among nonmoving bats should be much longer in duration (S2 Data). Second, foraging is likely to take up a substantial amount of the limited time outside the roost (mean = 2.4 hours). After commuting, searching, and selecting a host, a vampire bat can take up to 30 minutes to select a wound site, 10 to 40 minutes to prepare the wound site, and 9 to 40 minutes to feed [14,32]. Third, we used IR video to observe several interactions on or near cattle that were consistent with the short durations of foraging encounters in the proximity data (e.g., S1, S2 and S4 Videos). Fourth, foraging encounters among close female kin had a median duration of 9 seconds and were longer than among non-kin (median duration of 1 second), which is consistent with observations that affiliative interactions last longer.

## No clear evidence for highly coordinated collective movements

For animals with fluid social structures (e.g., high fission–fusion dynamics), it is important to clarify the ambiguous meaning of a "social group," and, similarly, one must distinguish

between different possible forms of "social foraging" [3]. In bats, the relative degree of social coordination during foraging can be difficult to assess and compare due to differing limitations in the observational methods and the lack of knowledge of differentiated social relationships within the colony. In this study, we took advantage of well-described within-roost relationships to assess evidence for several alternative scenarios of foraging behavior (Fig 1). Kinship and rates of association and cooperation led to longer and more frequent foraging encounters, but we did not observe highly coordinated joint departures or collective movements (Fig 1). This fluid pattern, of not moving in coordinated stable groups yet repeatedly encountering preferred associates during foraging, is also reflected in co-roosting networks where individuals form roosting groups that frequently change composition, yet maintain preferred relationships over time [16]. Given the many unsampled bats inside the same tree (approximately 200), it is possible that bats departed with other unobserved roostmates, but we did not see departures of large groups (while catching bats outside the roost) nor did we see evidence for coordination between roosting and departing in the tagged bats.

The ways that specific bats reunited with preferred associates therefore remain unknown, but the downward sweeping calls that we recorded in foraging bats (Fig 3) are similar to individually variable contact calls that vampire bats use to find and recognize preferred partners [29]. The role of calls, in particular a possibly foraging-specific call type ("n-shaped call" in Fig 3), warrants further investigation. In several other bat species, there is abundant evidence for socially influenced foraging based on eavesdropping on echolocation calls (e.g., [33–37]). The omnivorous greater spear-nosed bat in Trinidad appears to coordinate group foraging based on a group-specific contact call [38], and, in the fish-eating greater bulldog bat, female roostmates appear to depart individually, then assemble into small groups outside the roost to forage together, possibly coordinating their movements with calls [39].

## Affiliative and competitive interactions

Given the difficulty of making a bite compared to the ease of drinking from an open wound, some individual vampire bats appear to exploit the bites already made by others and fights can occur over open wounds or hosts [14,16,22,39,40], but it remains unclear how often these competitive interactions occur among familiar versus unfamiliar vampire bats. In our study, we observed foraging vampire bats engaging in both affiliative and competitive interactions (see Table C in S1 Text and S1–S5 Videos), and the competitive interactions were far more aggressive than what we observed among familiar captive bats feeding from an accessible and unlimited source of blood. This observation and our results above are consistent with the hypothesis that aggressive competitive interactions are more likely between less familiar bats.

The fluid nature of foraging encounters has potential implications for social dominance. Dominance hierarchies should be common when animals move together in groups, because the same frequent groupmates will also be primary competitors for first access to food [8,9]. Dominance hierarchies among familiar female vampire bats, which do not always travel or forage together, are indeed less clear and linear than among female mammals that do travel and forage in more stable groups [41]. Furthermore, blood from an open wound is not as limited of a resource as a discrete food item, so competition over food might be relatively low among familiar vampire bats that tolerate each other (as observed in captivity) and even share food, compared to unfamiliar conspecifics that might "steal" a wound.

Foraging behavior and social preferences may create a feedback loop. Social relationships can guide foraging decisions and help individuals gain access to defendable food [1,9]. For instance, experimental manipulation of social structure in zebra finches can impact how individuals forage together [42]. Conversely, decisions about where to forage may influence the

formation of social bonds. For instance, dolphins that share individual preferences for foraging sites or behaviors are more likely to associate in other contexts [2,43]. In vampire bats, stable isotope analyses suggest that individuals within the same colony have individualized foraging preferences; they repeatedly target different kinds of prey such as cattle versus sea lions [44]. As in dolphins, shared foraging preferences might similarly help drive social associations in vampire bats.

## Implications for cooperation

Vampire bats might benefit from foraging with socially tolerant partners (rather than alone or with random strangers) by acquiring social information on where to feed or by gaining access to open wounds. A single open wound can sequentially feed several bats, and allowing a close social partner to sequentially feed on the same open wound could be less costly to a successful forager than regurgitating blood to that individual later at the roost. Put differently, socially bonded bats could benefit from each other's foraging success, creating interdependence [45]. The presence of a socially bonded partner might even allow for joint defense of food against third parties, as seen in ravens [46,47].

Such forms of social foraging in vampire bats may have implications for the spatial scale of competition—a key factor shaping social evolution in humans [48] and other group-living animals [49]. In female vampire bats, cooperation occurs "locally" with specific frequent roostmates, and competition over food might occur more "globally" with members of the much larger population. If so, a more "global" scale of competition could reduce conflict and increase interdependence among highly associated females. To test this idea, it would be useful to determine if sampled groups of vampire bats consistently feed on the same or different prey individuals and if vampire bats are more likely to approach or avoid the social calls of foraging bats that are frequent roostmates versus unfamiliar conspecifics.

## Implications for describing social structure

A major advantage of proximity sensors is the ability to continuously track associations among multiple individual bats both inside and outside their roost, which allows for the construction of dynamic and multilayer networks. Studies on social foraging and other social behaviors in bats and other small highly mobile vertebrates have historically been limited by the available tracking technology [25]. Radiotelemetry has poor spatial resolution and continuously tracking many individuals is difficult. Current Global Positioning System (GPS) tags for bats have rather short runtimes, and the tags need to be recovered to download the data. Onboard ultrasound recorders (e.g., [34]) do not reveal the identity of encountered individuals. A major downside to proximity sensors was that many foraging encounters occurred at unknown locations. However, placing proximity sensors or antennas at more locations and on the livestock would allow a better reconstruction of foraging behavior. A combination of biologging approaches can also help to overcome existing challenges (e.g., [50,51]). Rapid standardized high-throughput methods for measuring social network structure, such as social proximity sensors, allow for social networks to be mapped quickly across multiple populations and species, enabling comparative studies investigating evolutionary and ecological drivers of social complexity across species.

## Supporting information

**S1 Text. Supporting information methods and results.**
(DOCX)

**S1 Fig. Time of foraging bouts by bat and day.**
(PDF)

**S1 Data. Data for Fig 2.**
(XLSX)

**S2 Data. Data for Fig B in S1 Text.**
(CSV)

**S3 Data. Data for Fig C in S1 Text.**
(XLSX)

**S4 Data. R-script for creating Fig D in S1 Text.**
(R)

**S5 Data. Data for Fig F in S1 Text.**
(CSV)

**S6 Data. Data for S1 Fig.**
(CSV)

**S1 Video. Three cows are grazing within few meters distance.** Each of the 3 cows has a vampire attached to its neck. Two of the bat individuals seem to be vocalizing in the direction of the other individuals (seconds 1 to 5 and 23 to 24).
(MP4)

**S2 Video. A vampire bat seems to be making a bite on the neck of a cow.** A second vampire bat joins and both engage in fight and fly away.
(MP4)

**S3 Video. One bat is drinking from an open wound on the neck of a cow.** The feet of a second bat hanging on the opposite site of the neck are visible. The first bat moves around, and both bats make body contact. The first bat gets hit by the ear of the cow, then both bats start pushing each other from one side of the cow neck to the other side, and a social call is audible (second 28; likely a "z"-call).
(MP4)

**S4 Video. Two bats feed from different wounds on the same cow.** The cow starts walking toward a second cow. One bat flies up and returns. When the first cow gets pushed by the second cow, the bats fly away.
(MP4)

**S5 Video. Two bats feed from different wounds on the same cow, and one bat vocalizes but not in the direction of the other bat.**
(MP4)

## Acknowledgments

We thank O. Castrellón and C. de León for permission to conduct fieldwork on their properties and D. Josic, J. Berrío-Martínez, V. Flores, M. Le Chevallier, B. Cassens, N. Duda, R. Crisp, M. Nowak, and G. Cohen for their assistance during field work. We are grateful to M. Knörnschild and A. Fernandez for supporting the collection and analysis of acoustic data, I. Waurick for her valuable assistance and expertise during molecular lab work, R. Crisp for observations of co-feeding, and I. Razik and E. Siebert for creating the line drawings in Fig 1 (I.R.: cattle and

tree; E.S.: bats). We thank D. Dechmann, J. Kohles, A. Fernandez, and J. Wilkinson for providing valuable feedback on earlier versions of this manuscript.

## Author Contributions

**Conceptualization:** Simon P. Ripperger, Gerald G. Carter.

**Data curation:** Simon P. Ripperger, Gerald G. Carter.

**Formal analysis:** Simon P. Ripperger, Gerald G. Carter.

**Funding acquisition:** Simon P. Ripperger, Gerald G. Carter.

**Investigation:** Simon P. Ripperger, Gerald G. Carter.

**Methodology:** Simon P. Ripperger, Gerald G. Carter.

**Project administration:** Simon P. Ripperger, Gerald G. Carter.

**Resources:** Gerald G. Carter.

**Supervision:** Gerald G. Carter.

**Validation:** Gerald G. Carter.

**Visualization:** Simon P. Ripperger, Gerald G. Carter.

**Writing – original draft:** Simon P. Ripperger, Gerald G. Carter.

**Writing – review & editing:** Simon P. Ripperger, Gerald G. Carter.

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
