## [Editor Report · Decision Letter 0]

30 Apr 2021

Dear Simon, 

Thank you for submitting your manuscript entitled "Social foraging in vampire bats is predicted by long-term cooperative relationships" for consideration as a Research Article by PLOS Biology.

Your manuscript has now been evaluated by the PLOS Biology editorial staff, as well as by an academic editor with relevant expertise, and I'm writing to let you know that we would like to send your submission out for external peer review.

IMPORTANT: We think that your paper would be better considered as a Short Report. It's already quite concise, so there's no need for any re-formatting at this stage, but please can you select "Short Reports" as the article type when you upload your additional metadata (see next paragraph)?

Please re-submit your manuscript within two working days, i.e. by May 04 2021 11:59PM.

Kind regards,

Roli

Roland Roberts

Senior Editor

PLOS Biology

rroberts@plos.org

---

## [Decision Letter · Decision Letter 1]

2 Jul 2021

Dear Simon,

Thank you very much for submitting your manuscript "Social foraging in vampire bats is predicted by long-term cooperative relationships" for consideration as a Short Report by PLOS Biology. As with all papers reviewed by the journal, yours was evaluated by the PLOS Biology editors as well as by an Academic Editor with relevant expertise and by three independent reviewers.

You'll see that the reviewers are broadly positive about your study, but each has a number of textual and/or presentational requests to improve the paper (plus some minor analyses). Based on the reviews, we will probably accept this manuscript for publication, provided you satisfactorily address the remaining points raised by the reviewers. Please also make sure to address the following data and other policy-related requests.

IMPORTANT:

a) Please attend to all of the requests from the reviewers. We note reviewer #2's suggestion that you remove the section (line 405) about calls; we actually think that this would be of interest to our readers, and if you decide to retain it, we suggest that you move one of the relevant (and rather interesting) supplementary Figs into the main main manuscript. Indeed, as a Short Report, you are allowed up to 4 main Figures, so if you feel that one of the other supplementary Figures would help address any of the clarity issues raised (especially for our broader readership, then do feel free to "promote" it to the main paper.

b) Please address my Data Policy requests further down. Essentially, we need the directly underlying data for some of the Figs to be provided, and for the location of the data to be cited in the legends.

We expect to receive your revised manuscript within two weeks. 

*Published Peer Review History*

*Early Version*

Sincerely,

Roli

Senior Editor,

rroberts@plos.org,

PLOS Biology

DATA POLICY:

We note that you present raw data and R scripts in the Figshare deposition. However, we also require the numerical values that underlie the figures and results of your paper be made available in one of the following forms:

Regardless of the method selected, please ensure that you provide the individual numerical values that underlie the summary data displayed in the following figure panels as they are essential for readers to assess your analysis and to reproduce it: Figs S2, S3, S4, S7. NOTE: the numerical data provided should include all replicates AND the way in which the plotted mean and errors were derived (it should not present only the mean/average values).

DATA NOT SHOWN?

REVIEWERS' COMMENTS:

Reviewer #1:

This paper is interesting, particularly in the use of proximity sensors in free-ranging bats. Overall, the data appears sound and the writing clear. 

Specific comments on the manuscript:

Line 89: change to "departing, following, or foraging"

Lines 101-102: state which species of vampire bat

Lines 139 and 141: If you recorded 277 co-feeding cases, why were only 201 use in the analyses?

Line 157: Can you give more details about mechanism behind the proximity sensors, or at least reference the supplemental material here.

Lines 211-212: what is meant by "visual light"? Light not visible to the cows? The bats?

Line 214: Is this a different roost from the one used for the other data collection? Please clarify.

Line 220: Subsequently

Line 252: What is meant by "hour bin" in this sentence?

Lines 352-354: Please reword/clarify this sentence. The variation was greater than expected?

Line 392: p = 0.063 is not significant... reword this sentence

Line 399: Please explain why this is only "weak" evidence, when your statistical results don't differ from what you consider stronger evidence for other comparisons above.

Line 502: "not a limited resource" seems to contradict what you say above and below about competition for wounds.

End of discussion: Include a concluding statement at the end.

Best of luck on your paper.

Reviewer #2:

This is an interesting topic, asking pertinent questions in a relevant and fascinating model system. An awful lot of work has gone into gathering these data. 

Figure 1 was particularly useful for explaining the framework. 

I do have some comments and suggestions. 

General comments:

(1) What are the effects of having those animals in captivity? What is the carry-over effect? How natural would they be expected to behave, given they have been fed for a long period of time under captive conditions?

(2) What is the accuracy of the proximity loggers?

(3) What is the time between the data was gathered in the published study (Ripperger et al 2019) and it being used in this study? What is the likely stability of these associations? If published in 2019, likely gathered some time before that.

(4) Mass of loggers. These are high as a % for flying animals. General level of acceptance is 5%. 6.9% is high. Could this have restricted their flight duration? And caused clustering due to reduced flight durations?

Line 184, Lines 233-256:

I struggled to follow everything in this section. Is it possible please to walk the reader a little slower through what was done here.

Line 325:

Out of interest, why do you think they wait so long before departing the roost site after dark?

Lines 326-329:

Is this a possible carry-over effect of captivity?

Line 330:

Should moon phase be controlled for in your analyses?

Lines 353-357:

Yes, greater than expected by chance. But what is the cattle had been highly clustered for various reasons, meaning the food source is highly concentrated (this is post cattle corralling). Is this just a product of food distribution?

Lines 378-386:

These are quite confusing for the reader to follow. Could you possibly elaborate here and provide more information and detail. 

Line 386-388:

"Tended". It's either significant or it is not. It is not.

Line 405:

This reads a little like an afterthought and add on. I would actually suggest removing this entirely from the paper, as it gets lost amongst all the other findings. It would work better as a stand-alone paper, presenting the novelty of this call. 

Discussion:

Figure 1 does a great job of visualising the logic behind these analyses and findings. I would suggest referring back to it to in the Discussion to keep the same structure and framework. 

Line 427:

What would be the benefit of foraging together? It would only be beneficial if individuals co-operate? Otherwise greater risk of prey becoming alert to presence?

Lines 462-466:

Is it just site preference that explains these findings? Certain individuals have preferred foraging areas, due to previous success there, and thus return? 

Line 469:

Over distance can these calls be heard? How close does an individual have to get to another individual to be able to hear these calls?

Discussion General Comments:

The Discussion does a good job of describing the results, but I don't think it really sets the results into any perspective, and is very narrow focused. It almost exclusively talks about the current study, with very little reference to prior work or work outside the immediate system. The limited reference to prior work is all bats. In contrast, the Introduction is broader and encompassing. As a consequence, the Introduction and Discussion feel quite disconnected. 

How do individuals recognise each other? It would be useful to have a few sentences describing this, for the benefit of non-specialist readers. 

I think the Latin names of species need adding where appropriate. 

Reviewer #3:

I very much enjoyed reading this manuscript, and I felt it fit the format of a 'Short Report' perfectly. Specifically, the research uses a range of new technologies and a charismatic system to shed new light on the relationship between social foraging and social relationships (particularly cooperative ones). As well as the broader implications, the findings also provide some novel new natural history type discoveries on this particular species too (which is great). I found that the writing was strong and concise, and that the methods and analyses were intuitive and suitable for providing these findings and the emerging conclusions drawn from them. As such, I only have minor comments on this manuscript (primarily just in relation to clarifying specific aspects of the report).

Abstract

Line 30: Change "controlling" to "accounting"

Line 31: The "and a previously undescribed call type" comes across as a bit 'out-of-the-blue' here. I think if readers first read the abstract before the full manuscript, they won't know what this means or what it is referring to at this point (I didn't). 

Introduction

I very much enjoyed the introduction and found it to be broadly appealing and span a great deal of systems. In some places, it felt like it could perhaps benefit on also making reference to other types of benefits/resources (rather than focussing on foraging), as actually a lot of the concepts here apply to any resource or need. For instance, Fleischmann et al. 2011 Current Biology (https://www.cell.com/current-biology/pdf/S0960-9822(13)00784-7.pdf) may be of interest to make reference to, as it considers how bats weigh up social relationships compared to access to preferred roosting sites (instead of focussing entirely on foraging). If the authors hope to keep the focus on foraging needs in the introduction (particularly given the nature of a 'short report' format), then perhaps it may just be beneficial to make some reference in the discussion to say that the concepts considered here actually extend beyond foraging, and future work could also be considered in relation to other needs/resources. 

Line 102: I don't think the phrase 'we used unpublished data' is needed here, as (1) I think the data will be published alongside the manuscript? And (2) I don't think previously-unpublished data should be viewed as innately more valuable than published/open data. 

Line 106-107: The final line of the introduction phrases the study a little too narrow and confined to a single social consideration within a single species, when actually I think the importance of these findings can be phrased in a much broader sense, as they give new general/fundamental insights into this phenomena in animal systems. 

Methods

Line 150-151: The MS states

"The results of this constrained permutation test and the unconstrained Mantel test were similar and gave the same conclusion so we report only the results from the double permutation test" 

But, given the amount of interest in these analytical techniques at the moment, and how standard models compare to those of null models, I would recommend including both sets of results (and placing one set in the supporting information for the more interested reader to refer to). 

Line 153: Why was a '1 hour' window chosen here? Were any other thresholds tried? 

Line 189-190: Change "see figure 2 in Ripperger et al. (2020b)" to "see Ripperger et al. (2020b)"

Results

Through the results, I'm unsure about which type of p values are being reported. Are these all the p values associated with the standard model parameters that are being reported? Or are these the p values derived from a separate permutation test against the standard model? If it is the latter, I recommend referring to it as 'P-null=' instead of just 'P='. If it is the former, then I think this needs specifying right at the start (and made clear in the methods). 

Line 356: The wording of '10 among 8' is a little confusing (sounds impossible at first notice), and this comes from this difficulty in the separation between foraging encounters and foraging pairs. I think this sentence and related phrases needs rephrasing for clarity. Perhaps it needs to be that each time a number is used, the associated term is also used e.g. instead of just saying '…10…', say '….10 foraging encounters…', etc.

L388: Why were one-tailed tests used here? Perhaps I missed this explanation in the methods, but would be good to make clear in results too briefly if using these. 

Discussion

Line 423-425: This second sentence of the discussion contains the key findings, but is currently a little complex to digest easily. I suggest breaking this into two sentences that follow on from one another. 

Line 483-503: I think that these two short (and separately sub-headed) paragraphs would be better if they were integrated into a single paragraph/subheading (as social dominance is captured broadly within competition anyway). 

Line 540-541: The final sentence of the discussion is phrased a little too methodological for an paper with such important fundamental findings. I think it would be more exciting to end the discussion with a sentence or so about the breadth of these findings and what they now suggest should be the future priorities/possibilities of interest.

Figure 1

Really found this figure very helpful to see exactly what was being tested in the manuscript. I suggest slightly changing the figure title from "…how within-roost relationships predict foraging…" to "…how within-roost relationships could predict foraging…"

Figure 2

Another great and useful figure. One question though, the legend states "To facilitate visual comparisons, we fixed the spatial coordinates of each node" but unclear how it was fixed? How were the positions in the circles defined initially, and which network were the positions fixed on? Or was it a 'best fit' position taken from average position over two separate networks?

I hope the authors find these minor comments useful for this interesting work.

Best wishes

Josh Firth (please note, I sign all of my reviews).

---

## [Editor Report · Decision Letter 2]

16 Jul 2021

Dear Simon,

On behalf of my colleagues and the Academic Editor, Catherine Hobaiter, I'm pleased to say that we can in principle offer to publish your Short Report "Social foraging in vampire bats is predicted by long-term cooperative relationships" in PLOS Biology, provided you address any remaining formatting and reporting issues. These will be detailed in an email that will follow this letter and that you will usually receive within 2-3 business days, during which time no action is required from you. Please note that we will not be able to formally accept your manuscript and schedule it for publication until you have made the required changes.

PRESS: We frequently collaborate with press offices. If your institution or institutions have a press office, please notify them about your upcoming paper at this point, to enable them to help maximise its impact. If the press office is planning to promote your findings, we would be grateful if they could coordinate with biologypress@plos.org. If you have not yet opted out of the early version process, we ask that you notify us immediately of any press plans so that we may do so on your behalf.

Best wishes,

Roli 

Roland G Roberts, PhD 

Senior Editor 

PLOS Biology

rroberts@plos.org